# Identification of a Novel Cuproptosis-Related Gene Signature for Prognostic Implication in Head and Neck Squamous Carcinomas

**DOI:** 10.3390/cancers14163986

**Published:** 2022-08-18

**Authors:** Shouyi Tang, Li Zhao, Xing-Bo Wu, Zhen Wang, Lu-Yao Cai, Dan Pan, Ying Li, Yu Zhou, Yingqiang Shen

**Affiliations:** 1State Key Laboratory of Oral Diseases, National Clinical Research Center for Oral Diseases, Chinese Academy of Medical Sciences Research Unit of Oral Carcinogenesis and Management, West China Hospital of Stomatology, Sichuan University, Chengdu 640041, China; 2State Institute of Drug/Medical Device Clinical Trial, West China Hospital of Stomatology, Chengdu 610041, China

**Keywords:** gene signature, head and neck neoplasms, overall survival, cell death

## Abstract

**Simple Summary:**

Head and neck squamous carcinoma (HNSC) is a common malignancy that requires novel therapeutic targets. Cuproptosis is an emerging research hotspot. The purpose of this study is to mine the cuproptosis-related genes to find prognosis-related genes. We successfully identified a 24-gene signature for predicting overall survival (OS) in HNSC patients and may expand the range of potential targets for treating HNSC.

**Abstract:**

Head and neck squamous carcinoma (HNSC) is a frequent and deadly malignancy that is challenging to manage. The existing treatment options have considerable efficacy limitations. Hence, the identification of new therapeutic targets and the development of efficacious treatments are urgent needs. Cuproptosis, a non-apoptotic programmed cell death caused by excess copper, has only very recently been discovered. The present study investigated the prognostic importance of genes involved in cuproptosis through the mRNA expression data and related clinical information of HNSC patients downloaded from public databases. Our results revealed that many cuproptosis-related genes were differentially expressed between normal and HNSC tissues in the TCGA cohort. Moreover, 39 differentially expressed genes were associated with the prognosis of HNSC patients. Then, a 24-gene signature was identified in the TCGA cohort utilizing the LASSO Cox regression model. HNSC expression data used for validation were obtained from the GEO database. Consequently, we divided patients into high- and low-risk groups based on the 24-gene signature. Furthermore, we demonstrated that the high-risk group had a worse prognosis when compared to the low-risk group. Additionally, significant differences were found between the two groups in metabolic pathways, immune microenvironment, etc. In conclusion, we found a cuproptosis-related gene signature that can be used effectively to predict OS in HNSC patients. Thus, targeting cuproptosis might be an alternative and promising strategy for HNSC patients.

## 1. Introduction

Until 2020, head and neck cancer remained one of the leading cancers worldwide, causing the rapidly growing burden of cancer incidence and mortality [1]. HNSC has rich diversity in primary sites and subtypes; thereby, the overall 5-year survival rate for HNSC remains low despite advances in cancer diagnosis and treatment. The ill effects, along with recurrence and metastasis, pose a tremendous financial burden and make optimal treatment and prognostic prediction crucial for HNSC patients [2]. A series of conventional and new prognostic signatures in HNSC were described, concerning immune-related genes [3], autophagy-related genes [4], microRNAs [5], oxidative stress-related genes [6], etc. However, more prognostic and therapeutic targets against HNSC are needed.

Copper-related biological functions have been under research and study for a long time. In eukaryotes, Cu acts as a redox cofactor for many copper-dependent enzymes and proteins [7]. It was first noticed in 1980 that Cu played a significant role in angiogenesis [8], dramatically inducing VEGF protein and its mRNA expression [9]. Like other trace metals essential to life, copper may lead to impairment and cell death if present at an inappropriate amount in cells. On the other hand, experts seek the possibility of harnessing it to kill tumor cells selectively, benefiting monotherapy. Cellular toxicity related to disrupted copper homeostasis involves various mechanisms, including accumulation of reactive oxygen species, proteasome inhibition, and anti-angiogenesis [10]. Moreover, a non-apoptotic program cell death caused by excess copper was recently termed “cuproptosis”. The disruption of lipoylated mitochondrial enzymes was recently found to be involved in this specific process [11].

With awareness of elevated copper concentration in tumor tissues and serum, the key role of Cu in the genesis, severity, and progression of cancer has attracted researchers’ interest. It could be a vulnerable point to target for arresting cancer development [12]. Attracted by the progress in copper-related cell death, we studied the application of copper in the field of tumors. Joint efforts are being made to achieve a state of copper deficiency or induce “cuproptosis” and seek potential strategies to attenuate the resistance and increase the selectivity of drugs [12,13,14]. An imbalance in copper homeostasis has also been observed in the progression of head and neck cancers [15]. In addition to cancer treatment, it is the foundation of application in predicting cancer risk and prognosis [16,17].

Here, we focused on whether those cuproptosis-related genes are associated with the prognosis of HNSC patients. In this study, we innovatively introduced cuproptosis-related genes into a prognostic multigene signature for HNSC, and we also performed further analysis to explore the underlying mechanisms. Conclusively, we made the first attempt to explore the unknown field with broad prospects, offering inspiration for more in-depth research related to cuproptosis in HNSC in the future.

## 2. Materials and Methods

### 2.1. Data Collection

The mRNA-sequencing data and related clinical information of 502 HNSC patients were downloaded from The Cancer Genome Atlas Program (TCGA) database (https://www.cancer.gov/about-nci/organization/ccg/research/structural-genomics/tcga, (accessed on 30 March 2022)). In addition, the microarray data profiles of GSE41613 and GSE42743 based on the same platform GPL570 and corresponding clinical information were obtained from Gene Expression Omnibus (GEO) database (https://www.ncbi.nlm.nih.gov/geo/, (accessed on 31 March 2022)). Then, the “sva” R package was used to merge the two datasets and remove batch effects. 

Derived from the genome-wide CRISPR-Cas9 loss-of-function test reported in previous literature [11], 347 potential cuproptosis-related genes (FDR < 0.05) were identified, and for a detailed list of genes, see Appendix A.

### 2.2. Development and Validation of a Prognostic Cuproptosis-Related Gene Signature

Differential expression of cuproptosis-related genes between normal and tumor samples was analyzed using the “limma” R package. Genes with a *p* value less than 0.05 were considered significantly different. TCGA cohort was classified as the training set, and the GEO cohort was the validation set. Then, we performed a univariate COX analysis of OS to identify which cuproptosis-related genes had prognostic values (*p* value < 0.05).

Mutations and associations of genes in the HNSC samples from the TCGA cohort were analyzed using the “maftools” R package. LASSO Cox regression analysis of prognostic-related genes was performed through the “glmnet” R package to construct a prognostic risk score model for OS prediction in HNSC patients. The model penalty parameter (λ) was determined by 10-fold cross-validations. The formula for calculating the risk score for each sample is defined as follows: risk score = e^sum (each gene’s expression × corresponding coefficient)^.

All samples were divided into low-risk and high-risk groups according to the median risk score of samples from the training set. Moreover, we used Kaplan–Meier analysis to compare the differences in OS between low- and high-risk groups. Then, we used the “timeROC” package to assess the accuracy of the model predictions. Subsequently, the test set was used to verify the reliability of the model.

### 2.3. Principal Component Analysis (PCA)

Aiming to figure out the distinction between the two groups, we used the “limma” R package to perform PCA on mRNA expression data before and after constructing the prognostic risk score model in the TCGA cohort. It should be pointed out that PCA was first performed on the expression profiles of cuproptosis-related genes, and the second PCA focused on genes in the risk score model.

### 2.4. Construction of a Nomogram for OS Prediction

A nomogram for predicting OS was built using the “rms” R package based on the gene expression profiles and clinical data of HNSC patients in the TCGA cohort. Time-dependent calibration curves and AUC curves were drawn to verify the validity of the nomogram. Univariate and multivariate Cox regression analyses were performed to assess if the prognostic risk score model could be an independent indicator for OS prediction of HNSC patients.

### 2.5. The Characteristic Distinction between the Low- and High-Risk Score Groups

The “limma” R package was adopted to determine the relationship between risk scores and clinical features, including age, gender, tumor grade, pathological stage, and AJCC TNM stage.

To compare immune-related characteristics between the two groups, immune cell infiltration and immune-related functions were quantified by ssGSEA analysis utilizing the “GSVA” R package. Gene sets used for the analysis were collected from previous studies [18,19]. Tumor Immune Dysfunction and Exclusion (TIDE) website (http://tide.dfci.harvard.edu/, (accessed on 10 April 2022)) was used to predict response to cancer immunotherapy (anti-PD1 and anti-CTLA4) in the two groups. On the other hand, we used the “pRRophetic” R package to predict the sensitivity toward chemotherapeutics from the gene expression data. The method worked by building statistical models from gene expression and drug sensitivity data in a very large panel of cancer cell lines, then applying these models to gene expression data from primary tumor biopsies [20]. 

### 2.6. Functional Enrichment Analysis

The symbol ID of each DEG (|logFC| > 1, *p*-value < 0.05) was converted to Entrez Gene ID using “org.Hs.eg.db” R package. Then, DEGs were subjected to Gene Ontology (GO) enrichment analysis and Kyoto Encyclopedia of Genes and Genomes (KEGG) analysis through the “clusterProfiler” R package.

### 2.7. Gene Set Variation Analysis (GSVA)

GSVA analysis of genetic maps was performed by the “GSVA” R package to compare the differences in biological pathways between low- and high-risk groups. The gene set “c2.cp.kegg.v7.4.symbols” from Gene Set Enrichment Analysis (GSEA) website (http://www.gsea-msigdb.org/gsea/msigdb/index.jsp, (accessed on 4 April 2022)) was used as the reference gene set.

### 2.8. PPI Network

We analyzed prognostic cuproptosis-related DEGs via the STRING website (https://cn.string-db.org/, (accessed on 8 April 2022)) to obtain the PPI network with an interaction score of more than 0.40 (median confidence). Further, the PPI network data were processed and displayed by Cytoscape software (version: 3.9.1, the Cytoscape Consortium, NY, USA, and the hub genes from prognostic cuproptosis-related DEGs were obtained by a Cytoscape extension called cytoHubba (version: 0.1, the Cytoscape Consortium, NY, USA).

### 2.9. Cell Culture

Six HNSC cell lines (UM1, SCC9, HN12, CAL27, H400, and SCC25) and the human normal squamous epithelial cell line (NOK) were used in this study. The UM1 cell line was obtained from the Cell Bank of the Chinese Academy of Science (Shanghai, China). The NOK, SCC9, HN12, CAL27, and SCC25 cell lines were purchased from American Type Culture Collection (ATCC; Manassas, VA, USA). The H400 cell line was established at Bristol Dental School, University of Bristol, Bristol, UK [21]. Identity authentication of all the cell lines was conducted by multiplex STR profiling. All the cell lines used in this study were stored in State Key Laboratory of Oral Diseases (West China Hospital of Stomatology, Sichuan University, Chengdu, China). UMI, HN12, and CAL27 cells were cultured in DMEM (Sigma, Saint Louis, MO, USA, D5546) supplemented with 10% FBS (HyClone, Logan, UT, USA, SH30088.03) and 1% penicillin–streptomycin solution (Gibco, Waltham, MA, USA, 15070063). SCC9, SCC25, and H400 cells were cultured in DMEM/F12 (Pricella, Greenacres, FL, USA, PM150310) supplemented with 10% FBS (HyClone, SH30088.03), 1% penicillin–streptomycin solution (Gibco, 15070063), and hydrocortisone (500 ng/mL; MCE, HY-N0583). NOK cells were grown in defined keratinocyte-SFM (Gibco, 10744019) supplemented with Defined Keratinocyte-SFM Growth Supplement (Gibco, 10744019) and 1% penicillin–streptomycin solution. All cultures were maintained in a humidified incubator with 5% CO_2_ at 37 °C.

### 2.10. RNA Isolation and RT-qPCR

Total RNA was isolated from cells using the Quick-RNA MiniPrep kit (Zymo research, Irvine, CA, USA, R1054). Then, reverse transcription was performed with the Takara PrimeScript RT reagent kit (Takara, Kusatsu, Japan, RR037A). Real-time quantitative PCR was performed with the ChamQ Universal SYBR qPCR Master Mix (Vazyme Biotech Co., Nanjing, China, Q711-02). All the experiments were performed according to the product’s instructions. Primers used in this study were designed via Primer 3 (https://primer3.ut.ee/, (accessed on 16 July 2022)). MRPS7 primer sequences were: forward primer 5′-AAGCCAGTGGAGGAGCTAA, reverse primer 5′-GCTTGATGGAAGATGGTGTA; GAPDH primer sequences were: forward primer 5′-GGAGCGAGATCCCTCCAAAAT, reverse primer 5′-GGCTGTTGTCATACTTCTCATGG. The ΔΔ*C*T value of the expression of MRPS7 was measured and assessed against the value of GAPDH.

### 2.11. Statistical Analysis

Wilcoxon rank-sum test was carried out to compare the difference between low- and high-risk groups. OS was compared by Kaplan–Meier analysis. Univariate and multivariate Cox regression analyses were used to obtain predictors of OS in HNSC. All statistical analyses were performed with R software (version: 4.1.2). If not specified, a *p* value < 0.05 was considered statistically significant.

## 3. Results

### 3.1. Identification of Prognostic Cuproptosis-Related DEGs in the TCGA Cohort

We found 238 differentially expressed cuproptosis-related genes between tumor and normal tissues in the TCGA cohort, and 39 of them were associated with prognosis (Figure 1a,b). Univariate Cox regression analysis revealed a total of 39 genes related to prognosis with a *p*-value less than 0.05 (Figure 1c). Based on the same screening criteria, we analyzed all 55,266 genes in the RNA-Seq. We found there were 2169 genes associated with prognosis, and 39 of these genes were cuproptosis-related genes. A further hypergeometric test showed that the 39 overlapping examples of these prognosis-related genes and all the cuproptosis-related genes were statistically significant, which means there is a significant correlation between cuproptosis-related genes and HNSC patients’ survival. It was worth mentioning that five of them were downregulated in tumor samples, including DLAT, MTF1, PDZD4, FDX1, and RPS25, and the rest of the genes were all upregulated. As shown, the correlation of the candidate genes was investigated (Figure 1d). The somatic mutation profile of the 39 prognostic cuproptosis-related genes was first summarized. In total, 66 of 506 HNSC samples experienced mutation of cuproptosis-related genes, with a frequency of 13.04% (Figure 1e). DOT1L had the highest mutation frequency, and six genes (including TRIM32, MRPL17, MTG1, COA6, COPZ1, and NUDF85) exhibited neither type of mutation in HNSC patients. Further analyses identified a mutation co-occurrence relationship between RPS25 and CDC27, RPS25 and DLAT, RPS25 and ABCE1, RPS25 and PSMB5, MRPS23 and STK11, MRPS23 and TAF6L, MRPL21 and ABCE1, TAF6L and STK11, PSMB5 and CDC27, PSMB5 and DLAT, PSMB5 and ABCE1, FDX1 and MTF1, RSL1D1 and CDC27, ABCE1 and CDC27, ABCE1 and DLAT, MTF1 and ATIC, and DLAT and CDC27 (Figure 1f).

### 3.2. Prognostic Risk Score Model Developed in the TCGA Cohort

To construct the prognostic risk score model, we performed a LASSO Cox regression analysis to reduce the number of genes. A 24-gene signature was identified according to the optimal value of λ (Figure 2a,b). The risk score of each sample was calculated by the formula mentioned in the Materials and Methods section.

Based on the median cut-off value, the patients in the TCGA cohort were divided into two groups: the high-risk group (*n* = 245) and the low-risk group (*n* = 246). The model effectively distinguished HNSC samples into low- and high-risk groups, with PCA indicating the discrepancy in the distribution of patients in different risk groups (Figure 2c,d).

### 3.3. The Relationship between Risk Score and Clinical Characteristics

The Kaplan–Meier survival analysis indicated that patients in the high-risk group had a reduced survival time compared with their low-risk counterparts (Figure 2f). Testing samples from the GEO cohort were also divided into two groups, high- (*n* = 65) and low-risk (*n* = 66), according to the median cut-off value. The outcome was consistent with the previous analysis (Figure 2g), which meant the prognostic robustness and practicability of the risk score model were verified. Furthermore, patients in the TCGA cohort and GEO cohort were divided into three groups (high-, medium-, and low-risk) separately for further overall survival analysis. Our results showed that there was still a significant difference in survival among the three groups (Appendix A). Moreover, time-dependent ROC curves were plotted separately for one year, three years, and five years to assess the predictive accuracy of the prognostic risk score model for OS. AUC reached 0.704 at one year, 0.737 at three years, and 0.698 at five years (Figure 2h). Then, the distribution of risk scores in age, gender, tumor grades, pathological stages, and AJCC TNM stages of samples in the training set was analyzed (Figure 3a). Although there were no statistically significant correlations between risk scores and age, gender, and tumor grades, risk scores were correlated with TMN stages and pathological stages to some extent. Patients of T3/T4 had higher risk scores than those of T1/T2 (*p* < 0.05). Patients of N3 had higher risk scores than those of lower N stages (*p* < 0.05). Regarding the pathological stage, risk scores also increased in stage III/IV patients compared with stage I patients (*p* < 0.05). We believe that the lack of completeness and timing updates of clinical information limited the ability to draw significant conclusions. Another ROC at five years was used to compare the predictive accuracy for OS of those variates, and the AUC of risk scores was the highest (Figure 2e). Subsequently, univariate and multivariate Cox regression analyses were conducted among those factors. Only risk scores and pathological stages were proven to be independent prognostic predictors for OS, and risk scores had a much higher hazard ratio (Figure 2i,j).

### 3.4. Development of a Nomogram for OS Prediction

A nomogram with age, gender, tumor grades, pathological stages, TMN stages, and risk scores was created to predict OS in HNSC patients from the TCGA cohort (Figure 3b). The deficiency in HNSC patients’ clinical records resulted in some shortages of the integrated scoring system. AUC demonstrated assurance of prediction for prognosis using the nomogram, which reached 0.728 at one year, 0.785 at three years, and 0.741 at five years (Figure 3c). The accuracy of the nomogram was superior to other variables, such as risk, age, gender, grade, and stage, providing a higher practical value in prognostic prediction. The calibration curves at one year, two years, and three years showed that the nomogram is reliable in predicting the OS of HNSC patients to a great extent (Figure 3d). Moreover, both the univariate and multivariate Cox regression analysis indicated that the risk score and stage are independent prognostic factors. (Figure 3e,f).

### 3.5. Immune-Related Characteristics in the Low- and High-Risk Score Groups

In order to further explore the potential relationship between the risk score and immune-related characteristics, immune cell infiltration and immune functions were quantified by ssGSEA analysis. Results showed that native B cells, plasma cells, CD8 T cells, T follicular helper cells, and regulatory T cells (Treg) were enriched in the low-risk group, and resting NK cells, M0/M2 macrophages, activated dendritic cells, and eosinophils were enriched in the high-risk group (Figure 4a). In addition, CCR, checkpoint, cytolytic activity, HLA, T cell co-inhibition and co-stimulation, and inflammation-promoting and type II IFN response had a lower score in the high-risk group (Figure 4c). In further analysis concentrating on checkpoints, we found that the majority of checkpoints were expressed at a higher level in patients of low risk, such as PDCD1 and CTLA4. Interestingly, there were a few exceptions, including CD44, CD276, NRP1, and TNFSF9 (Figure 4b). Mechanistically, immune checkpoint overexpression engenders an immune-suppressive tumor microenvironment, indicating that the low-risk group could respond better to immunotherapy. However, strangely, the low-risk group had a higher TIDE value. A higher value in TIDE analysis indicated a higher potential of tumor immune evasion, thus, it was less likely to benefit from anti-PD1/CTALA4 (Figure 4d). GSVA enrichment analysis was conducted to investigate the biological behaviors in the two groups, from which we learned that most metabolism pathways were enriched in the high-risk group (Figure 5g). Enrichment in the TCA cycle, oxidative phosphorylation, pyruvate metabolism, glycolysis, and gluconeogenesis may play an essential role in the poor overall survival of the high-risk group.

### 3.6. Response to Chemotherapy Drugs

Chemotherapy is one of the most common treatments for HNSC. Therefore, the sensitivity of HNSC patients in the TCGA cohort towards common chemotherapeutics was predicted using the “pRRophetic” R package. We found that half-maximal inhibitory concentration (IC50) of bleomycin, doxorubicin, and gemcitabine was lower in the high-risk group, and the most significant difference was the sensitivity to bleomycin (Figure 5a–f). The results implied that samples in the high-risk group were more sensitive to all three kinds of chemicals. Unfortunately, none of the three kinds of drugs are standard chemotherapies for HNSC, but each of them has therapeutic prospects. The cytotoxicity of bleomycin is greatly enhanced by electroporation as a component of electrochemotherapy (ECT), which is a topical, safe, and reproducible treatment for primary, metastatic, and radioresistant HNSC [22,23,24]. Doxorubicin, chemically or physically modified, is of great therapeutic value for HNSC [25,26], and the drug also has the potential for HNSC immunotherapy [3]. Gemcitabine has been proven to be an active agent against HNSC and can improve patient survival in nasopharyngeal carcinoma [27,28]. As a drug with potent radio-sensitization properties [29], gemcitabine in a combination of other chemicals is also applied for concurrent chemoradiotherapy of HNSC [30,31,32,33]. Regretfully, our results showed that there was not a significant correlation between the above-mentioned drug responses and risk scores in the GEO cohort (Appendix A). Further research is needed to confirm differences in drug sensitivity between the two groups.

Additionally, the mutation of TP53 is the most common genetic alteration in human cancers [34]. It is worth noting that patients with TP53 mutation had significantly higher risk scores than those with TP53 wild (Figure 5h).

### 3.7. Enrichment Analysis in the Low- and High-Risk Score Groups

GO and KEGG analyses were performed on the DEGs between the high-risk and low-risk groups to elucidate the biological functions and pathways related to the risk score. What could be clearly seen was that DEGs from the TCGA cohort were enriched in many immuno-biological processes (Figure 6a). More specifically, results of GO analysis indicated that DEGs participate in the B cell receptor signaling pathway, humoral-immune-mediated response by circulating immunoglobin, complement activation, humoral immune response, regulation of B cell activation, and recognition and engulfment of phagocytosis (Figure 6c). This could lead to the difference in immune response against cancer and response to treatment between the two groups. Moreover, KEGG analysis indicated that DEGs have a role in several signaling pathways and metabolic pathways (Figure 6b). Specifically, DEGs were enriched in cell adhesion molecules, estrogen signaling pathway, staphylococcus aureus infection, taurine and hypotaurine metabolism, α-linolenic acid metabolism, transcriptional misregulation in cancer, drug metabolism via cytochrome P450, and linoleic acid metabolism (Figure 6d). The results indicated the different biochemical activities between high- and low-risk groups.

### 3.8. Protein-Protein Interaction (PPI) Network of Prognostic Cuproptosis-Related DEGs

STRING online database was used to determine the PPI of cuproptosis-related DEGs, and Figure 6e presents the interaction of cuproptosis-related DEGs. Then, PPI network data were subjected to Cytoscape software to be further processed. To identify the hub genes, we applied the degree algorithm in the “cytoHubba” package of Cytoscape software. A total of 10 genes were identified, including MRPL15, MRPS7, MRPL21, MRPL17, MRPS23 and MRPS5, MTG1, RPS25, SRP54, and SSBP1 (Figure 6f).

MRPS7, MSPS5, MRPS23, MRPL15, MRPL21, and MRPL17 are members of mitochondrial ribosomal genes, encoding subunits of mitochondrial ribosome proteins (MRPs), which can help in protein synthesis inside mitochondria [35]. This implies that mitochondria may play a critical role in cuproptosis. Moreover, we performed further analysis focusing on MRPS7, the most central hub gene. Survival analysis indicated that higher mRNA expression of MRPS7 was strongly associated with poorer HNSC patients’ prognoses (Figure 7b). The specific difference in immune cell infiltration in the tumor microenvironment between patients with high and low expression of MRPS7 was explored. MRPS7 median expression value was used as the cut-off value. Our result showed that tumors with high expression of MRPS7 had significantly increased infiltration in activated NK cells and M2 macrophages when compared to those with low expression (Figure 7a). Connections with clinical features, including age, gender, tumor grades, clinical stages, and AJCC-TNM stages, were also analyzed. Notably, significant differences were observed in the expressions of MRPS7 between patients of a different gender. In addition, patients in grades 2/3 were more active in the expression of MRPS7 than those in grade 1. Significant differences also existed between patients in stage I and higher stages. As for TNM stages, patients in T3 had significantly higher expression in MRPS7 than in T1/2 (Figure 7c–f).

### 3.9. The Expression of MRPS7 in HNSC Cell Lines

In order to validate the expression of MRPS7 selected from the above analysis, we performed qPCR to measure the expression of MRPS7 in HNSC cell lines. Our result revealed that the expression of MRPS7 was significantly increased in SCC9, HN12, and H400 compared to the human normal squamous epithelial cell line (NOK). The expression of MRPS7 was slightly upregulated in CAL27 and SCC25. However, there was no difference in MRPS7 expression between UM1 and NOK cells (Figure 7g). In conclusion, MRPS7 was upregulated in various HNSC cell lines, and abnormal expression of MRPS7 might be associated with HNSC. It is worth noting that MRPS7 expression varies widely among different HNSC cell lines, suggesting potential heterogeneity between individual HNSC cell lines.

## 4. Discussion

In living matter, Cu exists in two oxidation states: cuprous (Cu^1+^) and cupric (Cu^2+^). Cu^2+^ is the main form in biological systems, for it is far more soluble than Cu^1+^, and Cu^1+^ is readily oxidized to Cu^2+^ in the presence of oxygen or other electron acceptors. The reversible process of oxidization gives Cu the ability to catalyze biochemical reactions. As a cofactor, copper supports the functions of diverse copper-dependent enzymes, including cytochrome-*c* oxidase (COX), superoxide dismutase 1 (Cu/Zn-SOD), ceruloplasmin (CP), and angiogenin [36]. Concentrating on the cellular level, copper plays a crucial role in the physiologic activities of mammals, such as redox chemistry, mitochondrial respiration, iron absorption, free radical scavenging, and elastin cross-linking [37].

Similar to other trace elements, copper homeostasis is tightly controlled by the regulation of copper uptake, transport, and excretion in the cell and within individual intracellular compartments. After crossing the plasma membrane via membrane surface receptors, either hCtr1 or DMT1, the intracellular transportation and distribution of copper are related to Cu chaperone proteins which deliver Cu to specific intracellular sites and target enzymes, preventing the presence of free Cu ions [38]. Though cellular compartmentalization exists, there is efficient communication between the organelles to maintain homeostasis. According to a previous study, COX19 was a communicating molecule to deliver the change in mitochondrial copper homeostasis to the Golgi-based copper transporter ATP7A [39]. The evidence from this study reveals the communication of the copper homeostasis in mitochondria and the copper secretory/export pathway, but the specific mechanism underlying this communication remains unclear. Moreover, thermodynamic mechanisms were reported concerning the distribution of cellular copper. Copper is transferred among related enzymes by exploiting gradients of increasing copper-binding affinity, and those with the highest affinity may participate in the arrangement of cellular copper [40].

Disruptions in the homeostasis of Cu can cause cell dysfunctions contributing to a wide range of diseases. Mechanistically, one of the copper-related toxicities, ROS-induced oxidative damage, has been thoroughly studied. It is thought to overwhelm body antioxidant systems and induce DNA damage, lipid peroxidation, protein modification, etc. Other possible mechanisms, such as the alteration of lipid metabolism, hepatic gene expression, activation of acidic sphingomyelinase (Asm), and release of ceramide and protein-metal interactions, have been summarized once [41]. Of note, existing research recognizes the critical role played by Cu in the development and progression of cancers. The pathways by which copper influences cancer development and progression have been outlined in previous papers, and conclusively, cancer proliferation, angiogenesis, and metastasis are key points during the interaction [42,43,44,45]. In addition, the serum copper and intracellular copper level of malignant tissues is elevated compared to healthy subjects, showing some connection with the onset and recurrence of cancers [46,47]. Another vital aspect of Cu is its role in anti-cancer treatment. Chelation, ionophore, and some other therapies are based on the focal target of copper [48]. As early as 1975, Schwartz had reviewed the role of trace elements, including copper, and underlined their potential roles as carcinogens and as diagnostic/prognostic markers [49]. Recent evidence suggests that the variation in copper metabolism makes copper-related proteins of use as cancer biomarkers [50]. A recent study elucidated how excess copper induces cell death on the molecular level. The results showed that copper-dependent death occurs through direct binding of copper to lipoylated components of the tricarboxylic acid (TCA) cycle, leading to lipoylated protein aggregation, subsequent iron-sulfur cluster protein loss, and ultimately cell death following proteotoxic stress [11]. The distinct type of cell death is termed “cuproptosis”, which has quickly become a research hotspot and provides extra inspiration for drug development as well as clinical indicator improvement. Research and studies in depth are necessary.

The heterogeneity of HNSC makes it critical to develop accurate prognostic indicators. The molecular prognostic markers are receiving more and more attention, and could be used as effective supplements to traditional clinicopathological parameters. To overcome tumor heterogeneity, multiple molecular markers are needed to better reflect the HNSC prognosis. So far, this is the first study to investigate the relationship between cuproptosis-related genes and HNSC. We have thoroughly examined the expression of 347 potential cuproptosis-related genes in HNSC samples and found that 39 of those genes are associated with the prognosis of HNSC patients. Then, a new prognostic risk score model integrating 24 cuproptosis-related genes was initially constructed. Our prognostic risk score model can be used to evaluate the roles of cuproptosis-related genes in the prognosis of HNSC patients comprehensively and identify potential cuproptosis-related therapeutic targets. Significant differences in survival of HNSC patients between low- and high-risk score groups were observed in both training and test sets. What is more, both univariate and multivariate Cox regression analysis revealed that the prognostic risk score model was an independent prognostic indicator. Furthermore, the potential application of this prognostic risk score model was assessed by a risk-assessment nomogram.

To provide deeper insight into the role played by the prognostic risk score model in HNSC, we performed some further analyses. Several lines of evidence suggest that copper and its transporter could affect the outcome of chemotherapy [51,52,53]. So, we predicted the sensitivity towards chemotherapeutics in both high- and low-risk groups. We found that patients in the high-risk group were more sensitive to bleomycin, doxorubicin, and gemcitabine, which might help in choosing appropriate drugs for chemotherapy. Additionally, previous studies have revealed the critical role of copper in immunity [54]. More importantly, it has previously been observed that intratumoral copper could regulate the expression of PD-L1 and influence tumor immune evasion, further affecting the efficacy of immunotherapy [55]. Therefore, we compared the immune microenvironment between the high- and low-risk groups and predicted their response to immunotherapy. Our results revealed that there were some differences in immune function and infiltration between low- and high-risk groups. An immunosuppressive tumor microenvironment was observed in patients in the low-risk group, indicating that patients in the low-risk group could have better efficacy of immunotherapy. However, interestingly, we found that patients in the low-risk group were less likely to benefit from anti-PD1/CTALA4 therapy using the TIDE website. A possible explanation for this might be that the tumor immune microenvironment is a complex admixture of various cellular components, including immune cells, cancer cells, and stroma. It is difficult to quantify the immune microenvironment precisely. More importantly, we demonstrated apparent differences in metabolic pathways between the two groups. The most notable was that patients in the high-risk group were more active in oxidative phosphorylation and TCA cycle pathways, both of which take place in the mitochondria. Given the critical role of mitochondria in the process of cuproptosis, this result is instructive. Because of the differences existing between the two groups, we further explored the different genes in the two groups. MRPS7 was identified to be the most essential gene. High expression of MRPS7 was associated with poor prognosis in HNSC patients. As one of the mitochondrial ribosomal genes, MRPS7 mutation was identified as a novel cause of mitochondrial respiratory chain (RC) dysfunction [56]. It has been demonstrated that cuproptosis occurs by direct binding of copper to lipoylated components of the TCA cycle [11]. The TCA cycle and the mitochondrial respiratory chain are closely related as part of the respiratory pathway complex. It is possible, therefore, that MRPS7 is involved in the copper-dependent cell death process. However, these results should be further explored.

Although this study provides unique data for constructing a novel cuproptosis-related gene signature for predicting OS in patients with HNSC, it has several limitations. First, the prognostic model’s construction and validation were based on retrospective data from public databases. Imperfections in information are partly responsible for unsatisfactory results. More comprehensive real-world data are needed to validate its clinical utility. Since many essential prognostic genes for HNSC might have been excluded, the intrinsic weakness of merely considering a single hallmark to construct a prognostic model was unavoidable. In addition, cuproptosis-related genes used in this study were identified through the genome-wide CRISPR-Cas9 loss-of-function test reported in the previous literature. The exact roles of these genes in the process of cuproptosis remain unclear and need further investigation.

## 5. Conclusions

In summary, this cuproptosis-related risk score model presented here is proven to be helpful in predicting prognosis and chemotherapeutic effectiveness, indicating that it can effectively guide clinical practice and allow for more personalized treatment strategies. Moreover, it not only may expand the range of potential therapeutic targets for the treatment of HNSC but can also offer great potential to uncover novel mechanisms of cuproptosis in HNSC.

## Figures and Tables

**Figure 1 cancers-14-03986-f001:**
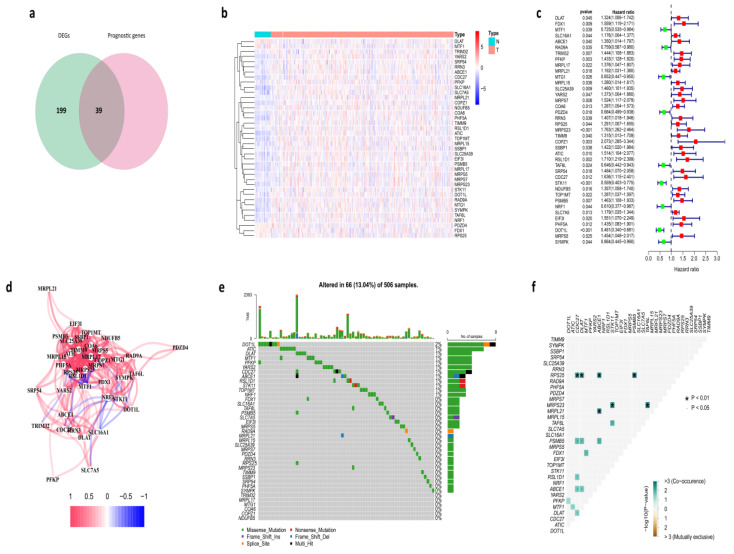
Identification of the prognostic cuproptosis-related genes in the TCGA cohort. (**a**) Identification of cuproptosis-related DEGs that were associated with OS by Venn diagram. (**b**) Heatmap of expression profiles of 39 prognosis-related genes. (**c**) The forest plot shows the results of the univariate Cox regression analysis. (**d**) The correlation network of candidate genes. (**e**) The mutation frequency of candidate genes in 506 HNSC samples from the TCGA cohort. (**f**) Co-occurrence and exclusion of mutation analyses for candidate genes.

**Figure 2 cancers-14-03986-f002:**
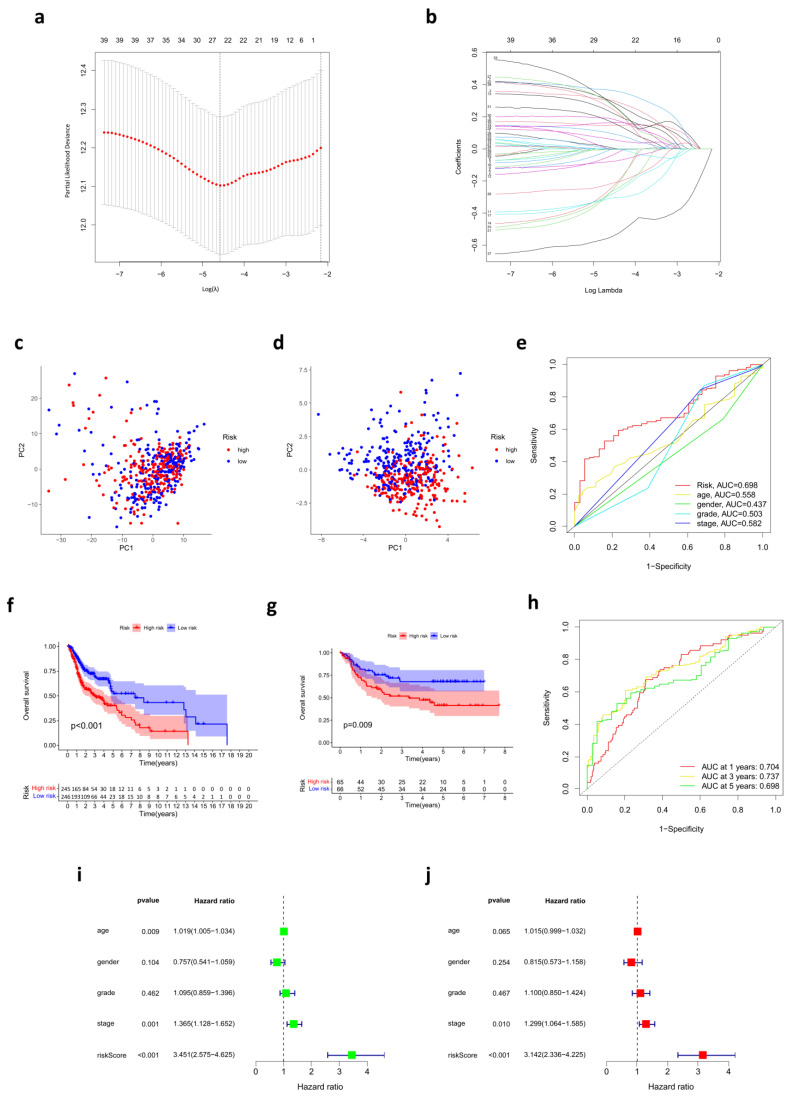
Prognostic risk score model established in the TCGA cohort. (**a**) Identification of genes used to build the prognostic risk score model. (**b**) LASSO coefficients of the 24 cuproptosis-related genes. (**c**) PCA based on all cuproptosis-related genes in HNSC. (**d**) PCA based on the genes involved in the prognostic risk score model in HNSC. (**e**) ROC curves for risk score and clinical features. (**f**,**g**) OS was compared between low- and high-risk score groups in the TCGA cohort (**f**) and GEO cohort (**g**), respectively. (**h**) Time-dependent ROC curves at separately one year, three years, and five years. (**i**,**j**) The forest plots show the results of univariate (**i**) and multivariate (**j**) Cox regression analysis in the TCGA cohort.

**Figure 3 cancers-14-03986-f003:**
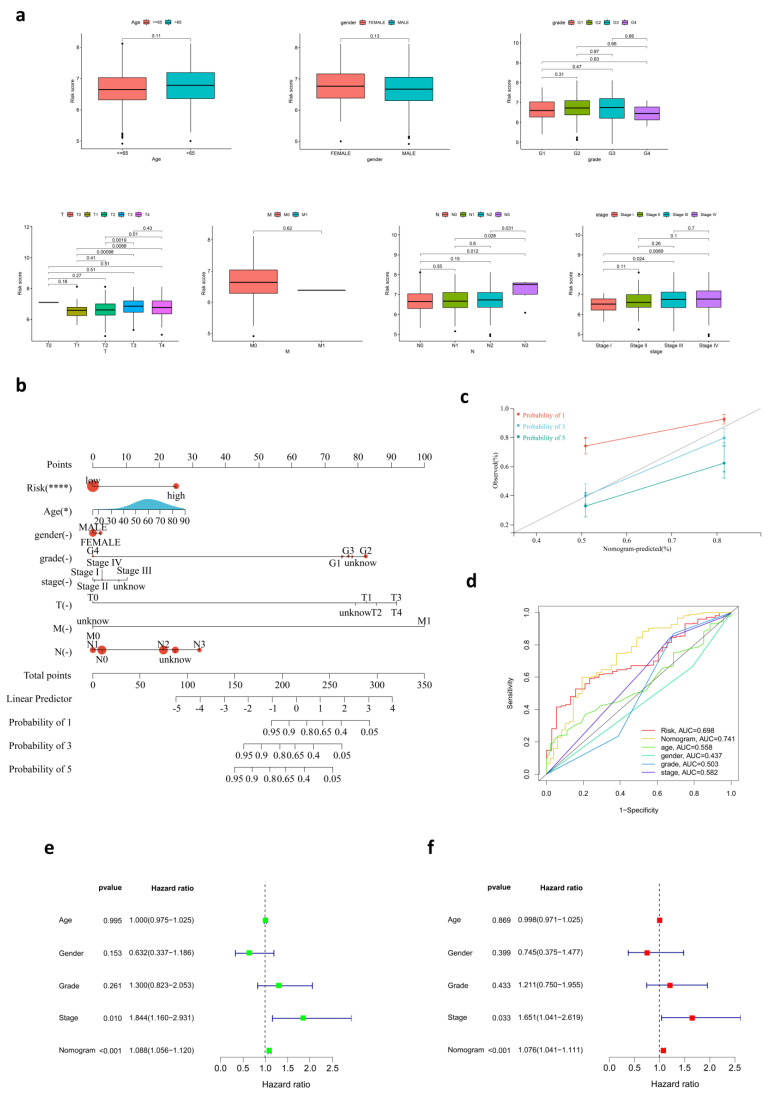
Development of a nomogram for OS prediction. (**a**) The relationship between risk score and clinicopathological variables, including age, gender, grade, stage, and TNM stage. (**b**) The nomogram for predicting survival in HNSC. (**c**) The calibration plots of the nomogram. (**d**) ROC curves for risk score, nomogram, and clinical pathological characteristics. (**e**,**f**) Univariate and multivariate Cox regression analysis for the nomogram and clinicopathological variables.

**Figure 4 cancers-14-03986-f004:**
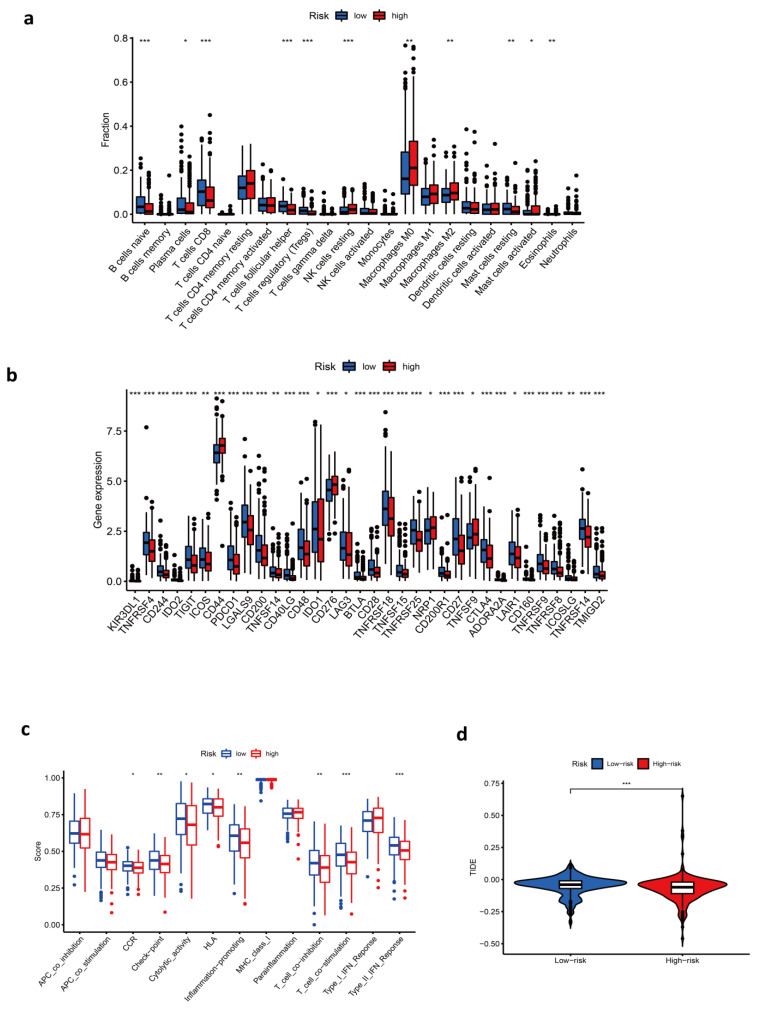
Immune-related characteristics in the low- and high-risk score groups. (**a**–**c**) Differences in immune cell infiltration (**a**), immune checkpoints (**b**), and immune functions (**c**) between low- and high-risk score groups. (**d**) Comparison of TIDE scores between low- and high-risk score groups. (***, *p* < 0.001; **, *p* = 0.001–0.01; *, *p* = 0.01–0.05.)

**Figure 5 cancers-14-03986-f005:**
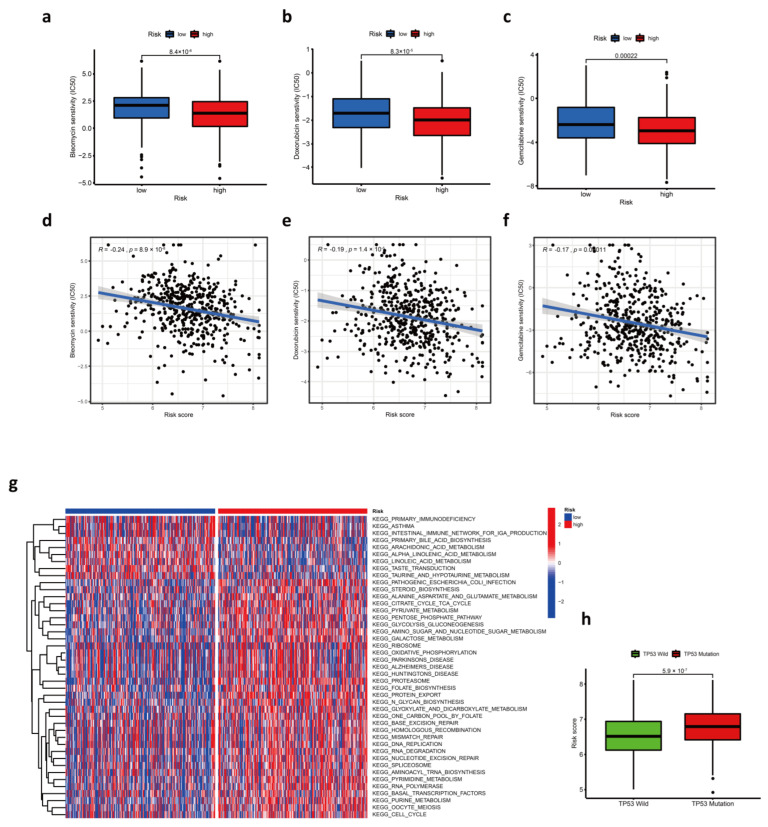
The role of prognostic risk score model in chemotherapy. (**a**–**c**) Comparison of estimated IC50 value of bleomycin (**a**), doxorubicin (**b**), and gemcitabine (**c**) between low- and high-risk score groups. (**d**–**f**) The correlation between risk scores and estimated IC50 value of bleomycin (**d**), doxorubicin (**e**), and gemcitabine (**f**). (**g**) The heatmap shows the result of GSVA enrichment analysis between low- and high-risk score groups. (**h**) Comparison of risk score between TP53 wild and TP53 mutation patients.

**Figure 6 cancers-14-03986-f006:**
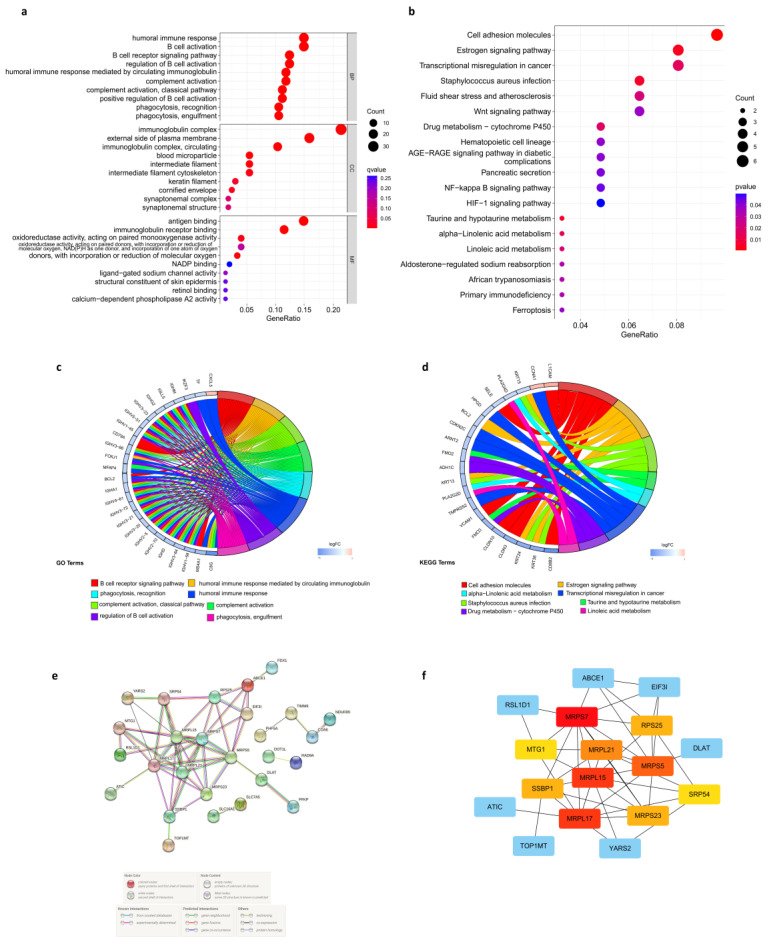
Enrichment analysis in the low- and high-risk score groups and PPI network. (**a**,**b**) GO and KEGG enrichment bubble charts. (**c**,**d**) String diagrams of GO and KEGG enrichment. (**e**) PPI network of cuproptosis-related DEGs. (**f**) Top 10 hub genes identified by cytoHubba.

**Figure 7 cancers-14-03986-f007:**
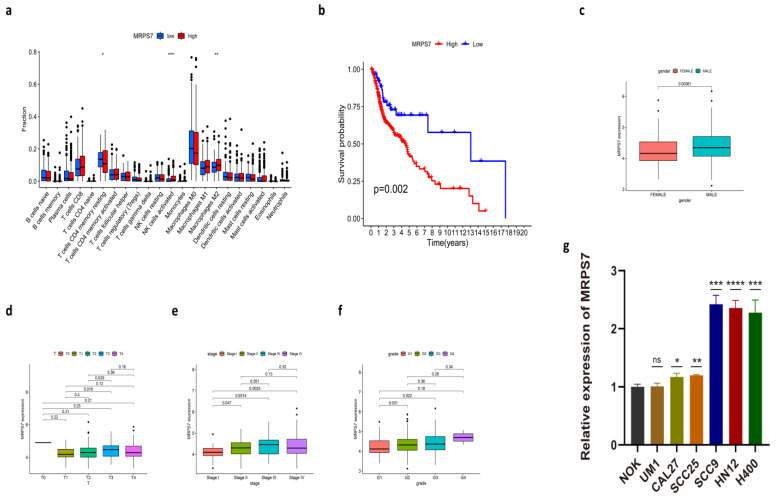
Further analysis focuses on MRPS7. (**a**) The difference in immune cell infiltration between low- and high-expression MRPS7 groups. (**b**) The comparison of OS between low- and high-expression MRPS7 groups. (**c**–**f**) The relationship between MRPS7 expression and clinicopathological variables, including gender (**c**), T stage (**d**), stage (**e**), and grade (**f**). (**g**) The expression of MRPS7 in NOK and HNSC cell lines. (****, *p* < 0.0001, ***, *p* < 0.001; **, *p* = 0.001–0.01; *, *p* = 0.01–0.05.)

## Data Availability

The data presented in this study are openly available in the TCGA database (https://www.cancer.gov/about-nci/organization/ccg/research/structural-genomics/tcga, (accessed on 30 March 2022)) and GEO database (https://www.ncbi.nlm.nih.gov/geo/, (accessed on 31 March 2022)).

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
