# Peer review of "Identification of a Novel Cuproptosis-Related Gene Signature for Prognostic Implication in Head and Neck Squamous Carcinomas"

_cancers, 2022, doi:10.3390/cancers14163986_

Round 1
Reviewer 1 Report
In this manuscript, Thang et. al picked an interesting and novel topic. The authors analyzed the following:
A. Identification of prognostic Cuproptosis-related DEGs in the TCGA cohort;
B. Prognostic risk score model developed in the TCGA cohort;
C. The relationship between risk score and clinical characteristics ;
D. Development of a nomogram for OS prediction ;
E. Immune-related characteristics in the low- and high-risk score groups;
F. Response to chemotherapy drugs;
G. Enrichment analysis in the low- and high-risk score groups;
H. Protein-protein interaction (PPI) network of prognostic cuproptosis-related DEGs.
However, I’d like to recommend some minor concerns for the manuscript as follows.
Major criticism:
1. The quality of figures is a big problem. Many words are too small to read, no matter how much I zoom in. This applies to almost all figures. The authors need to extensively optimize the font to make it into publishable quality. Make sure yourself read clear each single word before sending the revised version. Otherwise it is hard for the reviewers to do further evaluation.
2. After High-throughput screening, it is recommended to focus on certain hits to do further analysis. For example, is it possible to stain MRPS7 (or other genes of interest) using IHC on HNSCC tissue array vs normal tissue (or use other method to validate)?
Minor criticism:
1. There may be some issues with grammar. For example, line 46, “like” should be capitalized. Line 142 and line 143: “analysis” should be analyses. Line 152: consider inserting a comma before “and” to separate the elements, and similar issues are also elsewhere in the manuscript.
2. The references need to be slightly updated. For example, reference 4 from 1999 is too old.
3. I noticed the authors used two datasets in the GEO database as validation sets. But the authors did not indicate which method was used to combine the two datasets.
4. The result7: Enrichment analysis in the low- and high-risk score groups is generally clear, but the findings seem to be simple and superficial. its descriptive, generalizable character, should be strengthened.
5. According to figure5 d-f, the correlation coefficients of risk score and IC50 of bleomycin, doxorubicin, and gemcitabine were -0.24, -0.19, and -0.17, respectively. But the absolute values of correlation coefficients greater than 0.4 (r ≥0.4) are considered to be meaningful correlations usually. The authors do not point this out in the article.
Author Response
We are deeply thankful for your valuable comments that have greatly improved our manuscript. We presented the responses one by one to your comments as follows.
Major criticism:
1.The quality of figures is a big problem. Many words are too small to read, no matter how much I zoom in. This applies to almost all figures. The authors need to extensively optimize the font to make it into publishable quality. Make sure yourself read clear each single word before sending the revised version. Otherwise it is hard for the reviewers to do further evaluation.
Sorry for our error. By inserting figures in word context, they were automatically compressed, so we will upload figures with higher resolution in revised version.
2.After High-throughput screening, it is recommended to focus on certain hits to do further analysis. For example, is it possible to stain MRPS7 (or other genes of interest) using IHC on HNSCC tissue array vs normal tissue (or use other method to validate)?
Thank you for your kind opinion. In line with your suggestion, we performed an in vitro qPCR assay to validate the expression of MRPS7 in HNSC cell lines and human normal squamous epithelial cell line. And our results revealed that MRPS7 was upregulated in various HNSC cell lines.
Minor criticism:
1.There may be some issues with grammar. For example, line 46, “like” should be capitalized. Line 142 and line 143: “analysis” should be analyses. Line 152: consider inserting a comma before “and” to separate the elements, and similar issues are also elsewhere in the manuscript.
We have revised the grammatical issues according to your suggestion.
2.The references need to be slightly updated. For example, reference 4 from 1999 is too old.
We have updated the referencing accordingly and new references have been added.
3.I noticed the authors used two datasets in the GEO database as validation sets. But the authors did not indicate which method was used to combine the two datasets.
We used the “SVA” R package to merge the two GEO datasets and remove batch effects. And we have supplemented this point in the revised Materials and Methods.
4.The result7: Enrichment analysis in the low- and high-risk score groups is generally clear, but the findings seem to be simple and superficial. its descriptive, generalizable character, should be strengthened.
We have revised the text to address your concerns and hope that it is now clearer. Please see the lines 319-333 of the revised manuscript.
5.According to figure5 d-f, the correlation coefficients of risk score and IC50 of bleomycin, doxorubicin, and gemcitabine were -0.24, -0.19, and -0.17, respectively. But the absolute values of correlation coefficients greater than 0.4 (r ≥0.4) are considered to be meaningful correlations usually. The authors do not point this out in the article.
Thank you for the suggestions and opinions. Although the correlation coefficients of risk score and IC50 of bleomycin, doxorubicin, and gemcitabine were -0.24, -0.19, and -0.17 (the absolute values of which were all less than 0.4), the P values were all less than 0.05. So we think risk score and IC50 of bleomycin, doxorubicin, and gemcitabine were correlated.
We would like to thank the referee again for taking the time to review our manuscript.
Reviewer 2 Report
In the manuscript by Tang et al., the authors aim to evaluate the prognostic potential of the copper-induced cell death (cuproptosis) signatures in head and neck squamous carcinoma (HNSC). They measured risk scores based on the signatures and associated with patients’ overall survival in TCGA and other publicly available HNSC datasets. While the cuproptosis is an interesting mechanism as a novel cell death beyond apoptosis, the manuscript lacks sufficient logic and analysis to link the cuproptosis with clinical outcomes in HNSC. Several analyses they performed are not clearly justified without robust cross-validations and, moreover, many reports are simply descriptive without proper interpretation and described arbitrarily without robust cross-validations. I appreciate that the authors tried to explore many different features of tumors including immune activation potentials and chemo-sensitivity, but those analyses are not linked back to their main interest, cuproptosis, and they didn’t discuss how the cuproptosis are associated with their findings. More importantly, there are several other studies reporting prognosis signatures in HNSC, but the authors didn’t test whether their approach provide better prognostic signatures compared to others. Significant improvement with cross-validation is necessary for further consideration for publications. Major and minor comments are listed below.
Major comments:
1. Generally, submitted figures are not readable with very low resolutions. I can assume the characters on the figures based on the contexts, but they should replace the figures with high resolutions ones.
2. What is the justification for their investigation of cuproptosis in HNSC? As the authors stated in their Introduction, there are several prognostic factors reported recently. Their limitation and justification of improvement In the Introduction are critical to highlight the significance of the study.
3. Related to the above point, is there any previous report of copper over excess associated with Head and Neck cancer? If so, they should be mentioned in the Introduction.
4. The authors used a very loosened cutoff to call the significance as prognostic (p<0.05 from univariate Cox regression) and have about 1/6 of the cuproptosis-related genes as their candidate. To show that cuproptosis is associated with prognosis in HNSC, they should test all the genes using the same cutoff (p<0.05) and confirm whether significant genes are enriched within the cuproptosis-related genes. In this way, they can justify their interest in the prognostic potential of cuproptosis.
5. There are many papers that reported prognostic signatures in HNSC; https://doi.org/10.3389/fonc.2021.657002, https://doi.org/10.3389/fonc.2022.795781. They even used the same datasets and methods used in this manuscript (TCGA and the two GEO datasets) and reported 15- or 8 genes-based risk scores. The authors should compare their results with the previous results and show that the cuproptosis-based signatures provide better prognosis values than others for the same datasets.
6. Some method descriptions are too simple without the necessary information, so they need to be specified. For example, how the two GEO datasets are merged for survival analysis? Did the authors omit some samples for the analysis? If so, why and how? Where is the source of IC50 values for the chemotherapy drugs?
7. The authors used IC50 values of several drugs to associate with the risk scores of the patients. It is a bit confusing how the cell viability was measured for all TCGA patients. A clear description of the patients used in this analysis and methods to obtain IC50 values is missing.
8. The authors stratified into high and low-risk groups using the median value in TCGA and it does not make sense to use the same cutoff for the validation set which is very arbitrary. What is the distribution of risk scores in TCGA and the validation set? The authors should use the same stratification strategy for different independent datasets; use the median for each data. I also suggest grouping the patients into three or four groups with an equal number of patients in each group and comparing survival differences among them.
9. Hazard ratios of nomogram in Fig. 3 is almost close to 1 (upper CI ~ 1.1) and seem not strong enough to be a prognosis. What is the hazard ratio of the nomogram in the validation set?
10. How the risk score based on cuproptosis is associated with molecular subtypes of HNSC? This is a critical analysis to highlight the prognostic potential of the signature for specific subtypes since HNSC is highly heterogeneous. The authors may attempt to address this using Fig. 6, but the results are not well presented without any interpretations. A more comprehensive analysis associated with molecular subtypes would clarify the prognostic values of their signatures.
Minor comments:
1. On page 12, citations of Fig. 7a and Fig 7b need to be corrected.
Author Response
We are deeply thankful for your valuable comments that have greatly improved our manuscript. We presented the responses one by one to your comments as follows.
1.Generally, submitted figures are not readable with very low resolutions. I can assume the characters on the figures based on the contexts, but they should replace the figures with high resolutions ones.
Sorry for our error. By inserting figures in word context, they were automatically compressed, so we will upload figures with higher resolution in revised version.
2.What is the justification for their investigation of cuproptosis in HNSC? As the authors stated in their Introduction, there are several prognostic factors reported recently. Their limitation and justification of improvement In the Introduction are critical to highlight the significance of the study.
Thank you for your kind opinion. Cuproptosis is a very new area of research that caught our attention. And no research yet has integrated the role of cuproptosis in the development and progress of HNSC. More importantly, previous literature has revealed the critical role of copper in the development and progress of HNSC. So we want to explore the relationship between cuproptosis and HNSC.
As we stated in the Introduction, there are several prognostic risk score model reported recently. However, their research focuses were distinctly different. Previous studies focus on immune-related genes (doi:10.1016/j.csbj.2021.01.046), autophagy-related genes (doi:10.1111/jop.13231), microRNAs (doi:10.1016/j.heliyon.2020.e05436), and oxidative stress-related genes (doi:10.2174/1386207325666211207154436), etc. And no research yet focuses on cuproptosis-related genes.
3.Related to the above point, is there any previous report of copper over excess associated with Head and Neck cancer? If so, they should be mentioned in the Introduction.
Thank you for the most valuable comment. According to the previous literature, an imbalance in copper homeostasis has also been observed in the progression of head and neck cancers (DOI: 10.2174/0929867323666160405111543). And we have supplemented this point in the Introduction (lines 65-66 of the revised manuscript).
4.The authors used a very loosened cutoff to call the significance as prognostic (p<0.05 from univariate Cox regression) and have about 1/6 of the cuproptosis-related genes as their candidate. To show that cuproptosis is associated with prognosis in HNSC, they should test all the genes using the same cutoff (p<0.05) and confirm whether significant genes are enriched within the cuproptosis-related genes. In this way, they can justify their interest in the prognostic potential of cuproptosis.
Thank you for the very constructive comments. The aim of this study was to determine the relationship between cuproptosis-related genes and the prognosis of HNSC patients. We have attempted to do the analysis in accordance with your suggestion, but we found that there were many other genes affecting the results, which may mask the role of cuproptosis-related genes.
5.There are many papers that reported prognostic signatures in HNSC; https://doi.org/10.3389/fonc.2021.657002, https://doi.org/10.3389/fonc.2022.795781. They even used the same datasets and methods used in this manuscript (TCGA and the two GEO datasets) and reported 15- or 8 genes-based risk scores. The authors should compare their results with the previous results and show that the cuproptosis-based signatures provide better prognosis values than others for the same datasets.
Thank you for your instructive comment. As said before, although our study and other studies were based on the same dataset, our research focuses were distinctly different and all have prognostic value. Meanwhile, it is hard to prove that the cuproptosis-based signatures provide better prognosis values than others for the same datasets due to the lack of an accepted comparison program. What is important is that our prognostic risk score model have strong prognostic value. And we have thoroughly explored the relationship between our prognostic risk score model and HNSC.
6.Some method descriptions are too simple without the necessary information, so they need to be specified. For example, how the two GEO datasets are merged for survival analysis? Did the authors omit some samples for the analysis? If so, why and how? Where is the source of IC50 values for the chemotherapy drugs?
We used the “SVA” R package to merge the two GEO datasets and remove batch effects. And we have supplemented this point in the revised Materials and Methods (lines 83-84 of the revised manuscript).
We did not omit any samples during the analysis, expect when we evaluated the relationship between the risk sore and clinical features. We removed samples with missing clinical information.
We used “pRRophetic” R package to predict the sensitivity towards chemotherapeutics from the gene expression data. The method worked by building statistical models from gene expression and drug sensitivity data in a very large panel of cancer cell lines, then applying these models to gene expression data from primary tumor biopsies. For more details, please refer to this literature: pRRophetic: an R package for prediction of clinical chemotherapeutic response from tumor gene expression levels (DOI: 10.1371/journal.pone.0107468).
7.The authors used IC50 values of several drugs to associate with the risk scores of the patients. It is a bit confusing how the cell viability was measured for all TCGA patients. A clear description of the patients used in this analysis and methods to obtain IC50 values is missing.
As described in the last response, we used “pRRophetic” R package to predict the sensitivity towards chemotherapeutics from the gene expression data. The method worked by building statistical models from gene expression and drug sensitivity data in a very large panel of cancer cell lines, then applying these models to gene expression data from primary tumor biopsies. For more details, please refer to this literature: pRRophetic: an R package for prediction of clinical chemotherapeutic response from tumor gene expression levels (DOI: 10.1371/journal.pone.0107468).
8.The authors stratified into high and low-risk groups using the median value in TCGA and it does not make sense to use the same cutoff for the validation set which is very arbitrary. What is the distribution of risk scores in TCGA and the validation set? The authors should use the same stratification strategy for different independent datasets; use the median for each data. I also suggest grouping the patients into three or four groups with an equal number of patients in each group and comparing survival differences among them.
Thanks very much for this meticulous and precise suggestion. We divided the patients in the training set into high- and low-risk groups based on the median of risk scores. And it is a common practice to use the same cutoffs in the validation set. Otherwise, there will be no verification effect. Risk scores for all samples in the training set and validation set are available in Supplementary File 2. And we tried to divide patients into three of four groups, but the result is not satisfactory. We think it can better reflect the prognostic value of risk scores by dividing patients into two groups (high- and low-risk groups).
9.Hazard ratios of nomogram in Fig. 3 is almost close to 1 (upper CI ~ 1.1) and seem not strong enough to be a prognosis. What is the hazard ratio of the nomogram in the validation set?
Thank you for reading the paper so carefully. Although the hazard ratios of nomogram were not too high, the p value was extremely significantly (p<0.001). And the nomogram was developed by the data from the TCGA cohort, which has much richer clinical information. So we can not get the hazard ratio of the nomogram in the validation set.
10.How the risk score based on cuproptosis is associated with molecular subtypes of HNSC? This is a critical analysis to highlight the prognostic potential of the signature for specific subtypes since HNSC is highly heterogeneous. The authors may attempt to address this using Fig. 6, but the results are not well presented without any interpretations. A more comprehensive analysis associated with molecular subtypes would clarify the prognostic values of their signatures.
Thank you for the excellent suggestion. Actually, we have explored the classification of HNSC based on the risk score. Our results showed that the optimal cluster number was 2, which did not show the heterogeneity of HNSC well. So we did not add this part into the manuscript. We uploaded our analysis results to you.
Minor comments:
1.On page 12, citations of Fig. 7a and Fig 7b need to be corrected.
We have modified this mistake.
We would like to thank the referee again for taking the time to review our manuscript.

Reviewer 3 Report
The authors investigate potential cuproptosis-related genes in the publically available TCGA and GEO databases for head and neck cancer patients. They identify cuproptosis-related genes differentially expressed between tumor and normal samples in the databases. Overall, cuproptosis is an interesting new field for investigation. Use of these databases has its benefits, as well as limitations. I have suggestions for the authors:
1. The introduction could use a bit more discussion on cuproptosis and its potential impact in cancer treatment and cancer biology, given this is a fairly new discovery, and readers may be interested in more background.
2. Can the authors justify their set point for "high" and "low" risk cutoffs (lines 176-77) in more detail?
3. Can the authors elaborate on the "lack of completeness...of the clinical information" (lines 196-197)? How much clinical data was missing, and was this accounted for in the analyses?
4. Why did the authors include gender, age and grade in their nonogram when these variables did not reach significance in their univariable or multivariable analyses?
5. Can the authors discuss their biologic rationale in relation to cuproptosis underlying their investigation of immune-related characteristics and response to chemotherapy?
6. Bleomycin, gemcitabine and doxorubicin are not standard chemotherapies for head and neck cancer (which standardly uses platinum-based chemotherapy). Can the authors describe their rationale for their investigation and clinical relevance?
Author Response
We are deeply thankful for your valuable comments that have greatly improved our manuscript. We presented the responses one by one to your comments as follows.
1.The introduction could use a bit more discussion on cuproptosis and its potential impact in cancer treatment and cancer biology, given this is a fairly new discovery, and readers may be interested in more background.
Thank you for that valuable comment. We have revised the Introduction to address your comments and hope that it is now clearer. Please see the lines 59-68 of the revised manuscript.
2.Can the authors justify their set point for "high" and "low" risk cutoffs (lines 176-77) in more detail?
Thank you for this remark. Cutoffs were set at the median of risk scores of patients in the training set, which is a common practice in this type of research. And in the validation set, we divided the patients into high- and low-risk groups based on the same cutoffs. Further analysis showed that there was significant difference in survival between the two groups. This can also justify the rationality of the cutoffs.
3.Can the authors elaborate on the "lack of completeness...of the clinical information" (lines 196-197)? How much clinical data was missing, and was this accounted for in the analyses?
Thank you for your pointing it out. By this statement, we are meant to imply that the lack of a significant and close link between the risk score and certain clinical features of tumors partly attributes to the deficiency in completeness of the information available in the training set.
Grounded on our data of 528 items as total, the clinical information is somehow incomplete, in which TMN stages are missing in highest proportion (T stages: 23 missing; M stages missing: 271; N stages: 25 missing). Additionally, stages (75 missing), tumor grades (4 missing) and age (1 missing) are not fully recorded. We removed samples with missing clinical information when we evaluated the relationship between the risk sore and clinical features.
4.Why did the authors include gender, age and grade in their nonogram when these variables did not reach significance in their univariable or multivariable analyses?
Thank you for reading the paper so carefully. Usually, nomogram is defined as a graphic representation that consists of several lines marked off to scale and arranged in such a way that by using a straightedge to connect known values on two lines an unknown value can be read at the point of intersection with another line. In medical field, it is applied to diagnose or to predict disease onset or progression.
With multiple indicators combined, it could perform better in accuracy and validity. Though significant differences were not observed in previous analysis, the actual impact of these indicators on OS prediction is undeniable. Thus, we took age, gender and grade into account in the nomogram.
5.Can the authors discuss their biologic rationale in relation to cuproptosis underlying their investigation of immune-related characteristics and response to chemotherapy?
Thank you for your precious idea. We have added this aspect to the discussion according to your suggestion (see lines 442-444 and lines 447-451 of the revised manuscript).
6.Bleomycin, gemcitabine and doxorubicin are not standard chemotherapies for head and neck cancer (which standardly uses platinum-based chemotherapy). Can the authors describe their rationale for their investigation and clinical relevance?
Thank you for the very constructive comments. We evaluated the sensitivity of HNSC patients in the TCGA cohort toward common chemotherapeutics. And bleomycin, gemcitabine, and doxorubicin were the most meaningful results. Although none of the three kinds of drugs are standard chemotherapies for HNSC, there are many related studies to explore the relationship between these three kinds of drugs and HNSC. We thought these three kinds of drugs may have broad prospects in clinical application. And the detailed content was in lines 296-307 of the revised manuscript.
We would like to thank the referee again for taking the time to review our manuscript.
Round 2
Reviewer 1 Report
The authors addressed most of my concerns. However, the quality problem of many figures I raised last time has been resolved. This include but not limited to figure 1b,c,d,e, f; 5g, 6c, d, e ;7c-f(please check all other figures). These must be improved before re-consideration.
Author Response
Thank you again for taking the time to review the manuscript. The higher resolution of all figures were provided in the revised manuscript.
Reviewer 2 Report
The authors’ new manuscript has been somewhat improved from the previous version, but there are still significant flaws in the analysis with a lack of robustness. Two main concerns we have are 1) whether the cuproptosis-related genes are enriched in the survival-associated signatures; 2) whether the risk score-based stratification of HNSC patients separates good and poor prognostic populations in a robust way. These should be carefully addressed in the revised manuscript for further consideration. We put our comments after the author’s responses to our first comments below. New comments are marked with “->”.
1.Generally, submitted figures are not readable with very low resolutions. I can assume the characters on the figures based on the contexts, but they should replace the figures with high resolutions ones.
Sorry for our error. By inserting figures in word context, they were automatically compressed, so we will upload figures with higher resolution in revised version.
-> Several figures still have low resolutions. Even with a zoomed-in pdf file, texts are not readable in many panels. For example, Figure 1c, d, e. Fig 6c-e
4.The authors used a very loosened cutoff to call the significance as prognostic (p<0.05 from univariate Cox regression) and have about 1/6 of the cuproptosis-related genes as their candidate. To show that cuproptosis is associated with prognosis in HNSC, they should test all the genes using the same cutoff (p<0.05) and confirm whether significant genes are enriched within the cuproptosis-related genes. In this way, they can justify their interest in the prognostic potential of cuproptosis.
Thank you for the very constructive comments. The aim of this study was to determine the relationship between cuproptosis-related genes and the prognosis of HNSC patients. We have attempted to do the analysis in accordance with your suggestion, but we found that there were many other genes affecting the results, which may mask the role of cuproptosis-related genes.
-> The manuscript does not show whether there is a significant relationship between the cuprpotosis-related genes and the HNSC patients’ survival. Can they show whether the cuproptosis-related genes are significantly enriched in the survival signatures? As we stated before, about 1/6 of the cuproptosis-related genes satisfied their p-value (p<0.05). How many genes were significant (p<0.05) if they test all genes? If there is no significant overlap between all prognosis signatures and the cuproptosis-related genes, the authors’ hypothesis claiming prognostic values of the cuproptosis pathway couldn’t stand. This point is critical to justify the authors’ interest in the cuproptosis pathway in HNSC but is not still satisfied in this revised version.
5.There are many papers that reported prognostic signatures in HNSC; https://doi.org/10.3389/fonc.2021.657002, https://doi.org/10.3389/fonc.2022.795781. They even used the same datasets and methods used in this manuscript (TCGA and the two GEO datasets) and reported 15- or 8 genes-based risk scores. The authors should compare their results with the previous results and show that the cuproptosis-based signatures provide better prognosis values than others for the same datasets.
Thank you for your instructive comment. As said before, although our study and other studies were based on the same dataset, our research focuses were distinctly different and all have prognostic value. Meanwhile, it is hard to prove that the cuproptosis-based signatures provide better prognosis values than others for the same datasets due to the lack of an accepted comparison program. What is important is that our prognostic risk score model have strong prognostic value. And we have thoroughly explored the relationship between our prognostic risk score model and HNSC.
-> If the prognostic risk score model based on the cuproptosis-based signatures is not better or at least similar to other models, what is the significance of the model, then? The authors should discuss the unique and valuable features of the cuproptosis-based signatures compared to other previous signatures in the Discussion.
6.Some method descriptions are too simple without the necessary information, so they need to be specified. For example, how the two GEO datasets are merged for survival analysis? Did the authors omit some samples for the analysis? If so, why and how? Where is the source of IC50 values for the chemotherapy drugs?
We used the “SVA” R package to merge the two GEO datasets and remove batch effects. And we have supplemented this point in the revised Materials and Methods (lines 83-84 of the revised manuscript).
We did not omit any samples during the analysis, expect when we evaluated the relationship between the risk sore and clinical features. We removed samples with missing clinical information.
-> There are 200 samples from the two datasets, but it seems like only 153 patients are used in their analysis. Authors should describe how many samples from each GEO dataset are missed due to missing clinical information.
We used “pRRophetic” R package to predict the sensitivity towards chemotherapeutics from the gene expression data. The method worked by building statistical models from gene expression and drug sensitivity data in a very large panel of cancer cell lines, then applying these models to gene expression data from primary tumor biopsies. For more details, please refer to this literature: pRRophetic: an R package for prediction of clinical chemotherapeutic response from tumor gene expression levels (DOI: 10.1371/journal.pone.0107468).
-> This description should be included in Methods and the authors should mention that this information is “predicted” chemosensitivity not “actual” chemosensitivity in Results or Figure legends.
7.The authors used IC50 values of several drugs to associate with the risk scores of the patients. It is a bit confusing how the cell viability was measured for all TCGA patients. A clear description of the patients used in this analysis and methods to obtain IC50 values is missing.
As described in the last response, we used “pRRophetic” R package to predict the sensitivity towards chemotherapeutics from the gene expression data. The method worked by building statistical models from gene expression and drug sensitivity data in a very large panel of cancer cell lines, then applying these models to gene expression data from primary tumor biopsies. For more details, please refer to this literature: pRRophetic: an R package for prediction of clinical chemotherapeutic response from tumor gene expression levels (DOI: 10.1371/journal.pone.0107468).
-> Since the chemosensitivity can be measured based on gene expression, authors should estimate the sensitivity of samples from GEO to validate whether the drug responses are associated with risk scores.
8.The authors stratified into high and low-risk groups using the median value in TCGA and it does not make sense to use the same cutoff for the validation set which is very arbitrary. What is the distribution of risk scores in TCGA and the validation set? The authors should use the same stratification strategy for different independent datasets; use the median for each data. I also suggest grouping the patients into three or four groups with an equal number of patients in each group and comparing survival differences among them.
Thanks very much for this meticulous and precise suggestion. We divided the patients in the training set into high- and low-risk groups based on the median of risk scores. And it is a common practice to use the same cutoffs in the validation set. Otherwise, there will be no verification effect. Risk scores for all samples in the training set and validation set are available in Supplementary File 2. And we tried to divide patients into three of four groups, but the result is not satisfactory. We think it can better reflect the prognostic value of risk scores by dividing patients into two groups (high- and low-risk groups).
-> We disagree with the authors’ claim that it is a common practice to use the median value from the discovery set to separate patients in the validation set. The median is a relative value by its own definition. How can the absolute median value from the TCGA samples be used as the cutoff for the GEO patients? Authors should present the distribution of the risk score of the two datasets. If we use the GEO samples as primary and TCGA as the validation set, then the cutoff will be changed which may influence the results a lot. If authors want to keep binary classification, they use medians from each dataset.
And it is also hard to understand why results with 3 and 4 groups were not satisfactory when the multivariate results showed a significant Hazard ratio of the risk score in figure 2j. This brings a huge concern for the robustness of the prognostic value of the risk score.
Author Response
Thank you again for your comments and feedback. We would like to answer the questions you raised one by one below.
-> Several figures still have low resolutions. Even with a zoomed-in pdf file, texts are not readable in many panels. For example, Figure 1c, d, e. Fig 6c-e
All the figures have been changed to a higher resolution.
-> The manuscript does not show whether there is a significant relationship between the cuprpotosis-related genes and the HNSC patients’ survival. Can they show whether the cuproptosis-related genes are significantly enriched in the survival signatures? As we stated before, about 1/6 of the cuproptosis-related genes satisfied their p-value (p<0.05). How many genes were significant (p<0.05) if they test all genes? If there is no significant overlap between all prognosis signatures and the cuproptosis-related genes, the authors’ hypothesis claiming prognostic values of the cuproptosis pathway couldn’t stand. This point is critical to justify the authors’ interest in the cuproptosis pathway in HNSC but is not still satisfied in this revised version.
We thank you very much for your important comments. According to your suggestion, we have tested all genes based on the same criteria. Results showed that 2169 genes were associated with HNSC patients’ survival. And 39 genes were cuproptosis-related genes, accounting for about 1.8% of all these genes. Many published articles were also based on the same criteria (p<0.05 from univariate Cox regression) DOI: 10.1016/j.omto.2021.02.010, DOI: 10.1111/jop.13231, DOI:10.1016/j.csbj.2021.01.046, etc. Therefore, we believe that our results have revealed the significant relationship between the cuprpotosis-related genes and the HNSC patients’ prognosis.
-> If the prognostic risk score model based on the cuproptosis-based signatures is not better or at least similar to other models, what is the significance of the model, then? The authors should discuss the unique and valuable features of the cuproptosis-based signatures compared to other previous signatures in the Discussion.
Thank you for the suggestions and opinions. We addressed this point in the Discussion in line with your remarks (lines 444-448 and 452-455 of the revised manuscript).
-> There are 200 samples from the two datasets, but it seems like only 153 patients are used in their analysis. Authors should describe how many samples from each GEO dataset are missed due to missing clinical information.
We are sorry that we didn't explain it clearly. Following this comment, we double-checked the clinical data from the GEO database. And we have to admit that we made a mistake here. Actually, GSE41613 has 97 HNSC samples. Of these, 21 samples were excluded from the analysis because these patients did not die due to OSCC. GSE42743 has 74 HNSC samples and 29 normal samples. Of 74 HNSC samples, 19 samples were omitted from the analysis because these patients did not die as a result of HNSC. Therefore, a total of 131 HNSC samples were included in the analysis. Accordingly, we have also modified the corresponding content in the Results (lines 231-233 of the revised manuscript).
-> This description should be included in Methods and the authors should mention that this information is “predicted” chemosensitivity not “actual” chemosensitivity in Results or Figure legends.
Thank you for the suggestion. We have made corresponding modifications in the Methods and Results (lines 128-132 and line 302 of the revised manuscript).
-> Since the chemosensitivity can be measured based on gene expression, authors should estimate the sensitivity of samples from GEO to validate whether the drug responses are associated with risk scores.
Thank you for this suggestion. We did the same analysis in the GEO cohort. Regretfully, our results showed that there was not a significant correlation between the above-mentioned drug responses and risk scores in the GEO cohort (Supplementary Figure S1 a-f). And We have supplemented this point in the Results (lines 315-318 of the revised manuscript).
-> We disagree with the authors’ claim that it is a common practice to use the median value from the discovery set to separate patients in the validation set. The median is a relative value by its own definition. How can the absolute median value from the TCGA samples be used as the cutoff for the GEO patients? Authors should present the distribution of the risk score of the two datasets. If we use the GEO samples as primary and TCGA as the validation set, then the cutoff will be changed which may influence the results a lot. If authors want to keep binary classification, they use medians from each dataset.
And it is also hard to understand why results with 3 and 4 groups were not satisfactory when the multivariate results showed a significant Hazard ratio of the risk score in figure 2j. This brings a huge concern for the robustness of the prognostic value of the risk score.
Thank you for those comments. We re-analyzed the data in the GEO cohort by using the median risk scores of the GEO cohort as the cutoff. And respective sections have been modified accordingly (lines 231-233 of the revised manuscript). The distribution of the risk score of the two datasets was presented in Supplementary Figure S1g-h. As for the last question, we are very sorry that we misunderstood the review’s comments previously. Patients in the TCGA cohort and GEO cohort were divided into 3 groups for further overall survival analysis. Our results showed that there was still a significant difference in survival among the three groups (Supplementary Figure S1 j-k). We complemented this point in the revised manuscript (lines 235-238 of the revised manuscript).
Reviewer 3 Report
The authors have revised their manuscript appropriately. Only minor spelling and grammar issues. Otherwise, all my comments have been addressed adequately.
Author Response
Thank you again for taking the time to review the manuscript. We have modified the language to address your concern.
Round 3
Reviewer 1 Report
The authors addressed my comments and I have no further suggestion.
Author Response
Thank you very much.
Reviewer 2 Report
The revised version was improved a lot with more convincing data. Mainly, it is nice to see that the risk score-based stratification into either 2 or 3 groups showed significant survival differences in the two cohorts.
One last comment I have, it is still not clear whether the cuproptosis-related genes are enriched in prognosis in HNSC. How many genes did the authors test to get the 2169 survival-associated genes? Can the authors perform Fisher's exact test or hypergeometric test to show whether the 39 overlapping genes between 2169 and 347 are statistically significant or not? If it is significant, we can state that the cuproptosis has prognostic values. But if not, it is hard to justify focusing on the cuproptosis-related genes to find any prognostic potential signature.
Author Response
Again, we are appreciative of your comment. We have analyzed 55266 genes to get the 2169 prognosis-related genes. According to your suggestion, we performed the hypergeometric test using the “stata” R package. The code we used was as below: phyper(39-1, 2169, 55266-2169, 347, lower.tail=F). And the result showed that 39 overlapping genes between 2169 and 347 are statistically significant. We attached the screenshot of the hypergeometric test. And we have supplemented this point in the revised manuscript (lines 190-195 of the revised manuscript).
